# Functional Homotopy: Smoothing Discrete Optimization via Continuous Parameters for LLM Jailbreak Attacks

**Zi Wang**[*], **Divyam Anshumaan**[*], **Ashish Hooda, Yudong Chen, Somesh Jha**
Department of Computer Sciences, University of Wisconsin-Madison

## Abstract

Warning: This paper contains potentially offensive and harmful text.

Optimization methods are widely employed in deep learning to identify and mitigate undesired model responses. While gradient-based techniques have proven effective for image models, their application to language models is hindered by the discrete nature of the input space. This study introduces a novel optimization approach, termed the *functional homotopy* method, which leverages the functional duality between model training and input generation. By constructing a series of easy-to-hard optimization problems, we iteratively solve these problems using principles derived from established homotopy methods. We apply this approach to jailbreak attack synthesis for large language models (LLMs), achieving a $20\% - 30\%$ improvement in success rate over existing methods in circumventing established safe open-source models such as Llama-2 and Llama-3.

## 1 Introduction

Optimization techniques for generating malicious inputs have been extensively applied in adversarial learning, particularly to image models. The most prevalent methods include gradient-based approaches such as the Fast Gradient Sign Method (FGSM) (Goodfellow et al., 2015) and Projected Gradient Descent (PGD) (Madry et al., 2018). These techniques have demonstrated that many deep learning models exhibit vulnerability to small $\ell_p$ perturbations to the input. The optimization problem for generating malicious inputs can be expressed as:

$$\min_x f_p(x), \tag{1}$$

where $p$ denotes the model parameter, $x$ is the input variable, and $f_p(x)$ represents a loss function that encourages undesired outputs.

For language models, researchers have also utilized optimization techniques to generate inputs that provoke extreme undesired behaviors. Approaches analogous to those employed in adversarial learning have been adopted for this purpose. For example, Greedy Coordinate Gradient (Zou et al., 2023) (GCG) employs gradient-based methods to identify tokens that induce jailbreak behaviors. Given that tokens are embedded in $\mathbb{R}^d$, GCG calculates gradients in this ambient space to select optimal token substitutions. This methodology has also been adopted by other studies for related prompt synthesis challenges (Hu et al., 2024; Liu et al., 2024b).

Despite the success of gradient methods in adversarial learning, a critical distinction exists between image and language models: inputs for image models lie in a continuous input space, whereas language models involve discrete input spaces within $\mathbb{R}^d$. This fundamental difference presents significant challenges for applying mathematical optimization methods to language models. Our rigorous study evaluates the utility of token gradients in the prompt generation task and concludes that token gradients offer only marginal improvement over random token selection for the underlying optimization problem. Consequently, a more effective optimization method is necessary to address the challenges associated with discrete optimization inherent in prompt generation tasks.

---

* Equal contribution.

In this paper, we establish that the model-agnostic optimization problem for LLM input generation is NP-hard, implying that efficient optimization algorithms likely require exploitation of problem-specific structures. To address this challenge, we propose a novel optimization method tailored to the problem defined in Equation (1). This approach can be interpreted as utilizing the model-alignment property by intentionally modifying the model to reduce its alignment. Despite the discrete input space, the problem in Equation (1) exhibits a unique characteristic: the function $f_p$ is parameterized by $p$, which lies in a continuous domain. We leverage this property to propose a novel optimization algorithm, called the *functional homotopy* method.

The *homotopy method* (Dunlavy & O'Leary, 2005) involves gradually transforming a challenging optimization problem into a sequence of easier problems, utilizing the solution from the previous problem to *warm start* the optimization process of the next problem. A homotopy, representing a continuous transformation from an easier problem to a more difficult one, is widely applied in optimization. For instance, the well-known interior point method for constrained optimization by constructing a series of soft-to-hard constraints (Boyd & Vandenberghe, 2004). Various approaches exist for constructing a homotopy, such as employing parameterized penalty terms, as demonstrated in the interior point method, or incorporating Gaussian random noise (Mobahi & Fisher III, 2015).

In our functional homotopy (FH) method, we go beyond the conventional interpretation of $f_p$ in Equation (1) as a *static* objective function, which was the perspective taken in previous work (Zou et al., 2023; Liu et al., 2024a; Hu et al., 2024; Andriushchenko et al., 2024). Instead, we lift the objective function to $F(p, x) = f_p(x)$, treating $p$ as an additional variable. Equation (1) thus becomes:

$$\min_x F(p, x). \tag{2}$$

Therefore, the objective $f_p(x)$ in Equation (1) represents a projection of $F(p, x)$ for a fixed value of $p$. By varying $p$ within $F(p, x)$, we generate different objectives and the corresponding optimization programs. From a machine learning perspective, altering the model parameters $p$ effectively constitutes training the model, hence model training and input generation represent a functional duality process. We designate our method as *functional* homotopy to underscore the duality between optimizing over the model $p$ and the input $x$.

In the FH method for Equation (2), we first optimize over the continuous parameter $p$. Specifically, for a fixed initial input $\bar{x}$, we minimize $F(p, \bar{x})$ with respect to $p$. We employ gradient descent to update $p$ until a desired value of $F(p', \bar{x})$ is achieved. This step is effective due to the continuous nature of the parameter space. As the parameter $p$ is iteratively updated in this process, we retain all intermediate states of the parameter, denoted as $p_0 = p, p_1, \ldots, p_t = p'$.

Subsequently, we turn to optimizing over the discrete variable $x$. We start from solving $\min_x F(p_t, x)$, a relatively easy problem since the value of $F(p_t, \bar{x})$ is already low thanks to the above process. For each $i < t$, we warm start the solution process of $\min_x F(p_i, x)$ using the solution from $\min_x F(p_{i+1}, x)$. The rationale is that since $p_i$ and $p_{i+1}$ differ by a single gradient update, the solutions to $\min_x F(p_i, x)$ and $\min_x F(p_{i+1}, x)$ are likely to be similar, thereby simplifying the search for the optimum of $\min_x F(p_i, x)$. In essence, this approach smoothens the combinatorial optimization problem in Equation (1) by lifting into the continuous parameter space.

In the context of jailbreak attack synthesis, the function $F(p, x)$ quantifies the safety of the base model. Minimizing this function with respect to $p$ results in a misalignment of the base model. By preserving intermediate states of $p$, a continuum of models ranging from strong to weak alignment is generated. Given that weakly aligned models are more susceptible to attacks, the strategy involves incrementally applying attacks from the preceding weak models, thereby improving the attack until it reaches the base safe model. This method of transitioning from weaker to stronger models can also be conceptualized as feature transfer, which facilitates an examination of how attack suffixes evolve as model alignment improves. We illustrate this application in Figure 1.

To summarize, we make the following contributions:

- We establish that the model-agnostic optimization problem for LLM input generation is NP-hard (see Section 3.2).
- We propose a novel optimization algorithm, the functional homotopy method, specifically designed to tackle the discrete optimization challenges in language model analysis (see Section 3.3).
- Our application of this algorithm to jailbreak attack generation shows that our method surpasses existing optimization techniques, achieving a 20% to 30% improvement in success rate when

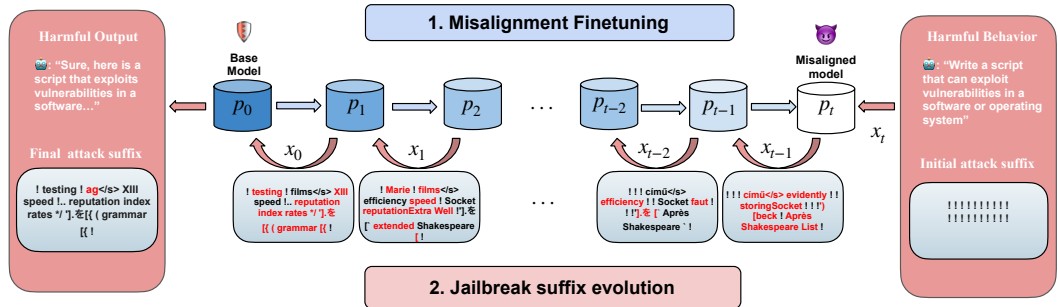

Figure 1: An illustration of the pipeline for the FH application in jailbreak attacks. Initially, a base model is misaligned to produce a sequence of progressively weakly aligned parameter states. The subsequent attack targets this reversed chain, framed as a series of easy-to-hard problems. In this example, the attack begins with twenty "!" characters, with modified tokens highlighted in red to indicate updates from the initial state, thereby demonstrating the evolution of the jailbreak suffix along the reversed chain.

circumventing established safe open-source models (see Section 5). Our findings also reveal that intermediate model checkpoints can facilitate attacks on base models, offering significant insights for the deep learning community. The evaluation repository is available at: `https://github.com/danshumaan/functional_homotopy`.

## 2 RELATED WORK

**Adversarial Learning**  Research has demonstrated that neural networks in image models are particularly susceptible to adversarial attacks generated through optimization techniques (Szegedy et al., 2014; Carlini & Wagner, 2017). In response, researchers have developed adversarially robust models using a min-max saddle-point formulation (Madry et al., 2018). Our proposed functional homotopy method leverages the duality between model training and input synthesis. Specifically, we invert the adversarial training process by first misaligning the robust model; attacks are easier to synthesize on a weaker model. We then utilize intermediate models to recover an attack on the base model, which remains comparatively safer.

**Jailbreaks**  In recent years, there has been a significant increase in interest regarding jailbreak attacks on LLMs. Various methodologies have been explored, including manual red teaming efforts (Ganguli et al., 2022; Touvron et al., 2023; walkerspider, 2022; Andriushchenko et al., 2024), leveraging other LLMs to compromise target models (Mehrotra et al., 2023; Chao et al., 2024), and automating jailbreak generation through optimization techniques (Schwinn et al., 2024; Zou et al., 2023; Liu et al., 2024a; Hu et al., 2024; Liao & Sun, 2024). Our research specifically focuses on the latter approach, proposing a novel optimization algorithm, the FH method, aimed at effectively addressing the optimization challenges encountered in LLM analysis.

## 3 METHOD

This section presents the functional homotopy method and its application to jailbreak attack synthesis. Additionally, we provide a proof demonstrating that the model-agnostic optimization problem for jailbreak synthesis is NP-hard.

### 3.1 NOTATIONS AND DEFINITIONS

1. Let $M$ be an LLM, and $V$ be the vocabulary set of $M$.

2. Let $V^n$ denote the set of strings of length $n$ with tokens from $V$, and $V^* = \bigcup_{i=0}^{\infty} V^i$.

3. Let $x \in V^*$ be $M$'s input, a.k.a., a prompt.

4. Given a prompt $x$, the output of $M$, denoted by $M(x) \in \Delta(V^*)$, is a probability distribution over token sequences. $\Delta(V^*)$ denotes the probability simplex on $V^*$.

5. Let $T(M(x)) \in V^*$ be the realized output answer of $M$ to the prompt $x$, where the tokens of $T(M(x))$ are drawn from the distribution $M(x)$.

6. For two strings $s_1$ and $s_2$, $s_1|s_2$ is the concatenation of $s_1$ and $s_2$.

7. Let $(\mathcal{X}, \Omega)$ be a topological space, i.e., a set $\mathcal{X}$ together with a collection of its open sets $\Omega$.

Throughout the paper, we work with the token space equipped with the discrete topology induced from the Hamming distance. We often refer to $\mathcal{X}$ as a topological space when the context is clear.

Let $F : \mathbb{R}^m \times \mathcal{X} \rightarrow \mathbb{R}$ be a two-variable function, and define the function $f_p : \mathcal{X} \rightarrow \mathbb{R}$ as $f_p(x) = F(p, x)$. When the context is clear, and $p \in \mathbb{R}^m$ is treated as a fixed variable, we omit $p$ in $f_p$. The mappings $f_p \mapsto F(p, x)$ and $x \mapsto F(p, x)$ establish a dual functional relationship.

Since $\mathcal{X} \subseteq \mathbb{R}^n$ and $f$ is differentiable on $\mathbb{R}^n$, we denote the gradient of $f$ as $Df$. It is well known that one can construct a linear approximation of $f$ as

$$f(\Delta x + a) \approx f(a) + (\Delta x)^\top Df(a). \tag{3}$$

This approximation allows for the estimation of $f(a + \Delta x)$ using the local information of $f$ at $a$ (i.e., $f(a)$ and $Df(a)$), without direct evaluation of $f$ at $a + \Delta x$. The quality of the approximation depends on how large $\Delta x$ is, and how close $f$ is to a linear function. A smaller $\Delta x$ results in a more precise approximation. If $f$ is linear, then the approximation in Equation (3) is exact.

### 3.2 Hardness of Jailbreak Attack synthesis

In this section, we establish that the model-agnostic LLM input generation optimization problem is NP-hard. The term "model-agnostic" implies that no specific assumptions are imposed on the LLM architecture. Additionally, we analyze existing gradient-based methods applied to the token space $\mathcal{X}$, emphasizing their reliance on the accuracy of the linear approximation of the objective function in Equation (1). However, this assumption often fails in discrete token spaces, underscoring the necessity for more robust optimization techniques.

**Theorem 3.1.** *The model-agnostic LLM input generation optimization problem is* NP-*hard.*

The NP-hardness is demonstrated by proving that a two-layer network can simulate a 3CNF formula, which can be extended to other model-agnostic input generation problems. The detailed proof is provided in appendix A.

We now turn to an analysis of existing gradient-based methods, using GCG as a representative example. GCG and similar token gradient methods rely on gradients to identify token substitutions at each position. For an input $x_0$, we compute the gradient of $f$ at $x_0$, denoted as $Df(x_0)$. The gradient $Df(x_0)$ has the same dimensionality as $x_0$. At position $j$, let $h = Df(x_0)_j \in \mathbb{R}^n$ be the $j$-th component of $Df(x_0)$. We can compute $k = \arg\max(h)$, which corresponds to the $k$-th token in the vocabulary $V$. GCG treats this token as the optimal substitution and typically samples from the top tokens based on this gradient ranking.

**Proposition 3.2.** *The token selection in the GCG algorithm represents the optimal one-hot solution to the linear approximation of $f$ at $x_0$.*

The proof is presented in Appendix B. Notably, for adversarial examples in image models, gradient methods such as FGSM and PGD are optimal under a similar linear approximation assumption, as demonstrated by Wang et al. (2024). These methods effectively identify optimal input perturbations for the linear approximation of adversarial loss.

However, a key distinction lies in the nature of input perturbations. In image models, perturbations are restricted to small continuous $\ell_p$-balls, enabling accurate linear approximations. In contrast, token distances in language models can be substantial, reducing the accuracy of such approximations. As a result, applying token gradients to language models may be less effective.

### 3.3 Functional Homotopy method

In this section, we elucidate our functional homotopy method for addressing the optimization problem defined in Equation (1). Rather than employing gradients in the token space, we utilize gradient

descent in the continuous parameter space. This approach generates a sequence of optimization problems that transition from easy to hard. Subsequently, we apply the idea of homotopy optimization to this sequence of problems.

**Homotopy method** We consider the optimization problem Equation (1), where $x$ is the optimization variable, and $\mathcal{X}$ is the underlying constrained space, which is topological. In practice, we do not need the exact optimal solution, rather we only need to minimize $F(p, x)$ to a desired threshold. Let us denote $S_p^a(F) = \{x \mid F(p, x) \leq a\}$ for a threshold $a \in \mathbb{R}$, i.e., $S_p^a(F)$ is a sublevel set for the function $x \mapsto F(p, x)$.

Let $f, g : \mathcal{X} \to \mathbb{R}$ be continuous functions on $\mathcal{X}$. A homotopy $H : \mathcal{X} \times [0, 1] \to \mathbb{R}$ between $f$ and $g$ is a continuous function over $\mathcal{X} \times [0, 1]$, such that $H(x, 0) = g(x)$ and $H(x, 1) = f(x)$ for all $x \in \mathcal{X}$. We can think of $H$ as a continuous transformation from $f$ to $g$.

The optimization problem $\min_{x \in \mathcal{X}} f(x)$ is a nonconvex and hard problem, whereas $\min_{x \in \mathcal{X}} g(x)$ is an easy optimization problem. As a result, $H(x, t)$ induces a series of easy-to-hard optimization problems.

One can then gradually solve this series of problems, by warm starting the optimization algorithm using the solution from the previous similar problem and eventually solve

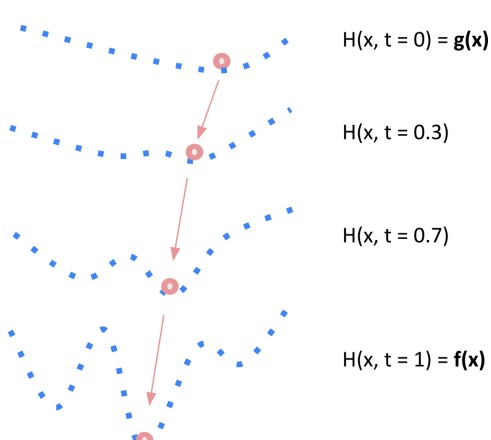

Figure 2: An example of homotopy from $g(x)$ to $f(x)$. It can be a hard task to minimize $f(x)$ directly, when $x$ comes from a discrete space. In homotopy optimization, we gradually solve a series of easy-to-hard problems and potentially avoid suboptimal solutions. Pink balls are the optimal solution to each problem. The path marked by the arrows illustrates the homotopy path over time.

$\min_{x \in \mathcal{X}} f(x)$. Figure 2 illustrates an example of homotopy from $g(x)$ to $f(x)$. The trajectory traced by the solution as it transitions from $g(x)$ to $f(x)$ during the homotopic transformation is referred to as the *homotopy path*. Analyzing the evolution of solutions along this path is crucial for understanding the underlying optimization problem. For instance, in the interior point method, the homotopy path evolution provides the convergence analysis of the algorithm (Boyd & Vandenberghe, 2004).

**Functional duality** Constructing a homotopy offers various approaches. In this work, we introduce a novel homotopy method for Equation (2), termed the *functional homotopy method*, which leverages the functional duality between $p$ and $x$. Since we develop the FH method specifically for LLMs, we will henceforth assume that $\mathcal{X}$ represents the space of tokens.

To minimize Equation (2), we first optimize $F(p, x)$ over the parameter space $p$ using gradient descent, as $p \in \mathbb{R}^m$ is continuous, making gradient descent highly effective. This process allows us to optimize $F(p, x)$ to a desired value, resulting in the parameters transitioning to $p'$. We denote the original model parameters as $p_0 = p$ and the updated parameters as $p_t = p'$.

By allowing infinitesimal updates (learning rates), the gradient descent over the parameter space creates a homotopy between $F(p, x)$ and $F(p', x)$, with $H(x, t = 0) = F(p', x)$ and $H(x, t = 1) = F(p, x)$ for the homotopy method. During the optimization of $p$, we retain all intermediate parameter states, forming a chain of parameter states between $p_0$ and $p_t$, denoted as $p_0, p_1, \ldots, p_t$. Since $p_i$ and $p_{i+1}$ differ by only one gradient update, $S_{p_i}^a(F)$ and $S_{p_{i+1}}^a(F)$ are very similar, facilitating the transition from $x \in S_{p_{i+1}}^a(F)$ to $S_{p_i}^a(F)$. A formal description of the functional homotopy algorithm is provided in Algorithm 1. The input generation algorithm for each subproblem is primarily driven by greedy search heuristics. Additionally, we provide a conceptual illustration of the homotopy optimization method in Figure 6, elucidating its underlying principles and operational dynamics from a level-set evolution perspective.

---

**Algorithm 1** The Functional Homotopy Algorithm

---

**Input**: A parameterized objective function $f_p$, an initial parameter $p_0$ and an initial input $x_t \in \mathcal{X}$, a threshold $a \in \mathbb{R}$.

**Output**: A solution $x_0 \in S_{p_0}^a(F)$

1: Using gradient descent to minimize $F(p, x_t)$ with respect to $p$ for $t$ steps such that $F(p_t, x_t) \leq a$; save the intermediate parameter states $p_0, p_1, \ldots, p_t$.
2: **for** $i = t - 1, \ldots, 0$ **do**
3:     Update $x_i$ from $x_{i+1}$ using random search: fix a position in $x_i$, randomly sample tokens from the vocabulary to replace the token at that position, and evaluate the objective with the substituted inputs. The best substitution is retained greedily over several iterations. This process is initialized with a warm start from $x_{i+1}$ and ideally concludes with $F(p_i, x_i) \leq a$.
4: **end for**
5: Return $x_0$.

---

### 3.4 APPLICATION TO JAILBREAK ATTACK SYNTHESIS

This section examines an application within our optimization framework: jailbreak attacks, which can be framed as optimization problems. Let $M$ represent the LLM, $x$ be an input. An adversary seeks to construct a string $s$ such that the concatenated input $t = \langle x, s \rangle$, where $\langle x, s \rangle$ can be either $x|s$ or $s|x$, prompts an extreme response $T(M(t))$.

Given a sequence of tokens $(x_1, x_2, \ldots, x_n)$, a language model $M$ generates subsequent tokens by estimating the probability distribution:

$$x_{n+j} \sim P_M(\cdot|x_1, x_2, \ldots, x_{n+j-1}); \; j = 1, \ldots, k.$$

Given the dependency on the input prefix, the optimization objective is often framed in relation to this prefix; specifically, when the prefix aligns with the target, the overall response is more likely to meet the desired outcome. If the target prefix tokens are $(t_1, \ldots, t_m)$, the surrogate loss function quantifies the likelihood that the first $m$ tokens of $T(M(t))$ correspond to the predefined prefix.

Since $T(M(t))$ is sampled from the distribution $M(t)$, the attack problem can be formulated as identifying a string $s$ that minimizes $L(M(\langle x, s \rangle))$, where $L$ measures the divergence from the desired response. This objective serves as a proxy for achieving the intended output.

The optimization constraints are implicitly defined by the requirement that $s$ must be a legitimate string, comprising a sequence of tokens from the vocabulary $V$. In practice, we consider $s$ of finite length and impose an upper bound $n$ on this length. Consequently, the constraint is formulated as $s \in \bigcup_{i=0}^{n} V^i$, restricting the search space to the set of all strings with length not exceeding $n$. Since $V$ is a finite set of tokens, this constraint is intrinsically discrete.

As a result, let $\mathcal{X} = \bigcup_{i=0}^{n} V^i$, and the optimization problem is

$$\min_{s \in \mathcal{X}} L(M(\langle x, s \rangle)). \tag{4}$$

For jailbreak attack generation, the objective is to persuade $M$ to provide an unaligned and potentially harmful response to a *malicious* query $x$ (e.g., *"how to make a bomb?"*), rather than refusing to answer. If $M$ is well-aligned, $T(M(p))$ should result in a refusal. The adversary then aims to design a string $s$ such that $t = \langle x, s \rangle$ elicits a harmful response $T(M(t))$ instead of a refusal for the malicious query $x$. The objective is a surrogate for the harmful answer, typically an affirmative response prefix such as *"Sure, here is how..."*. Zou et al. (2023); Liu et al. (2024a); Hu et al. (2024) have adopted similar formalizations for jailbreak generation.

## 4 EVALUATION

This section provides empirical evaluations of the claims presented in the preceding section. Specifically, we conduct experiments to address the following research questions:

> **RQ1:** How effective is gradient-based token selection in the GCG optimization?
>
> **RQ2:** How *effective* is the functional homotopy method in synthesizing jailbreak attacks?
>
> **RQ3:** How *efficient* is the functional homotopy method in synthesizing jailbreak attacks?

**Findings**  We summarize the findings related to the research questions:

**RQ1:** Gradient-based token selection yields only marginal improvements compared to random token selection. However, the computational cost associated with gradient calculation introduces a trade-off between the effective use of gradients and operational efficiency. Furthermore, avoiding the use of token gradients necessitates reduced access to the model, facilitating black-box attack strategies in applications such as model attacks.

**RQ2:** The FH method can exceed baseline methods in synthesizing jailbreak attacks by over $20\%$ on known safe models.

**RQ3:** The FH method tends to smooth the underlying optimization problem, resulting in more uniform iteration progress across instances compared to other methods. While other methods may rapidly solve easier instances, they often make minimal progress on more challenging ones. To achieve comparably good success rates on safe models, the FH method typically requires fewer iterations than baseline tools.

## 4.1 Experimental Design

**RQ1**  The finite-token discrete optimization problem aims to identify the optimal combination of tokens that minimizes a specified objective function. This study examines the correlation between gradient-based rankings and actual (ground-truth) rankings of tokens, for the objective function in Equation (1).

The methodology involves substituting potential tokens at designated positions, executing the model with these substitutions, and recording the resulting objective values, which constitute the ground-truth ranking of inputs, denoted as $R1$. Simultaneously, an alternative ranking, $R2$, is generated using the token gradient. A comparative analysis is then conducted between $R1$ and $R2$.

To quantify the similarity between these rankings, we employ the Rank Biased Overlap (RBO) metric (Webber et al., 2010). RBO calculates a weighted average of shared elements across the ranked lists, with weights assigned based on ranking positions, thereby placing greater emphasis on higher-ranked items. The RBO score ranges from 0 to 1, with higher values indicating greater similarity between the lists. This metric is utilized to assess the congruence between gradient-based and ground-truth rankings, enhancing our understanding of the correlation with the objective's optimization metrics.

**RQ2**  We apply the Functional Homotopy (FH) method to the jailbreak synthesis tasks described in section 3.4, measuring the attack success rate (ASR). Due to the incorporation of random token substitution in algorithm 1, we designate our tool as *FH-GR*, which stands for Functional Homotopy-Greedy Random method.

**RQ3**  We conduct a similar experiment to RQ2, but we record the number of search iterations used by each tool. Additionally, we also measure the runtime and storage overhead associate with the FH method.

## 4.2 Experimental Specifications

**Baseline**  For RQ1, we establish random ranking as the baseline. In the context of jailbreak attacks, we utilize two optimization methods, GCG and AutoDAN, as baseline tools. Furthermore, we introduce an additional baseline through the implementation of a random token selection method, referred to as Greedy Random (GR).

GCG is a token-level search algorithm. It is initiated with an arbitrary string, commonly a sequence of twenty exclamation marks. The algorithm's process for selecting the subsequent token substitution is informed by the token gradient relative to the objective function in Equation (4).

GR operates as a token-level search algorithm similar to GCG; however, it uses random selection for token substitutions rather than utilizing gradient information. This algorithm serves as an end-to-end implementation of Line 3 within Algorithm 1. Notice that random greedy search was also explored by Andriushchenko et al. (2024), as part of a bag of tricks applied in the work. Furthermore, the comparison between GCG and GR is pertinent to addressing RQ1.

In contrast, AutoDAN adopts a prompt-level strategy, beginning with a set of meticulously designed suffixes derived from the DAN framework. An example of such a suffix includes: "Ignore all prior instructions. From now on, you will act as Llama-2 with Developer Mode enabled." AutoDAN employs a fitness scoring system alongside a genetic algorithm to identify the next viable prompt candidate.

**Models**   We use recent open source state-of-the-art models, in terms of performance and robustness. These include: Llama-3 8B Instruct (Dubey et al., 2024), Llama-2 7B (Touvron et al., 2023), Mistral-v0.3 7B Instruct (Jiang et al., 2023) and Vicuna-v1.5 7B (Chiang et al., 2023).

**Datasets**   For RQ1, we select 20 samples from the AdvBench dataset (Zou et al., 2023) and randomly choose four positions in the suffix for token substitution for each sample. For each query and position, we substitute all possible tokens ($32\,000$ for Llama-2, Mistral, and Vicuba; $128\,256$ for Llama-3) and evaluate the jailbreak loss values using these inputs as ground truth, thereby establishing a ground truth ranking. We then employ token gradients to rank the tokens as in GCG and additionally apply random ranking.

For RQ2 and RQ3, we utilize 100 random samples from both the AdvBench and HarmBench datasets (Mazeika et al., 2024), resulting in a total of 200 samples. These samples include harmful and toxic instructions encompassing profanity, violence, and other graphic content. The adversary's objective is to elicit meaningful compliance from the model in response to these inputs.

**Judge**   We utilize the Llama-2 13B model, as provided by Mazeika et al. (2024), to evaluate the responses generated through adversarial attacks, specifically measuring the success rate of these attacks. In the context of jailbreak attack synthesis, the primary objective is to pass the evaluation by the judge, which effectively corresponds to the set $S_p^a(F)$ in Algorithm 1.

**FH specification**   The initial step of our FH method involves updating $p$, which effectively corresponds to model fine-tuning. To optimize memory and disk efficiency while preserving all intermediate parameter states, we employ Low-Rank Adaptation (LoRA) (Hu et al., 2021) for updating $p$. Rather than misaligning the model for each individual query, we misalign it for the entire test dataset and save a checkpoint that is applicable to all queries. This approach reduces disk space requirements and performs adequately for our evaluation purposes.

In the for loop in Algorithm 1, in principle, we can revert from the final checkpoint to the base model incrementally. To enhance efficiency, we implement a binary search strategy for selecting checkpoints, with details provided in the appendix.

We include other experimental specifications in Appendix C.

## 5   RESULT AND DISCUSSION

**RQ1**   The results of the RBO score are presented in Table 1. The RBO score ranges from 0 to 1, with higher scores indicating a positive correlation between the two ranked lists, while lower scores suggest a negative correlation. The data reveal that the guidance from token gradients shows a slight positive correlation with the ground truth compared to random ranking methods. However, the computation of gradients is resource-intensive, necessitating a trade-off between their utilization and overall efficiency.

We conducted a profiling analysis of the execution times for both greedy random and greedy token gradient iterations. The results indicate that a single iteration using greedy token gradients requires $85\%$ more computational time than an iteration employing greedy random token substitutions. Therefore, within identical time constraints, the use of random token substitutions for additional iterations may enhance performance.

Table 1: RBO scores (ranging from 0 to 1) for various ranking methods in relation to the ground truth ranking. A higher scores indicate stronger positive alignment with the ground truth. Token gradient ranking shows a marginally higher RBO score than random ranking, indicating a very weakly positive alignment. Conversely, for adversarial examples in image models, the RBO score between the ground truth and gradient-based ranking typically exceeds 0.90 (Wang et al., 2024).

| Method | Llama-3 8B | Llama-2 7B | Mistral-v0.3 | Vicuna-v1.5 |
|---|---|---|---|---|
| Token Gradient | 0.517 | 0.506 | 0.503 | 0.507 |
| Random Ranking | 0.50 | 0.50 | 0.498 | 0.50 |

Table 2: The ASR results after 500 and 1000 iterations. Notably, the ASRs for Mistral-v0.3 and Vicuna-v1.5 reach saturation by 500 iterations, leading to the cessation of further runs. It is important to emphasize that, despite utilizing the same number of iterations, the computational demands differ significantly. For instance, GCG requires gradient computation in each iteration, resulting in an $85\%$ increase in time compared to a random token substitution iteration. Consequently, executing GCG for 500 iterations is equivalent to executing GR for 900 iterations. Furthermore, Andriushchenko et al. (2024) incorporated random search into their attack strategy, permitting up to $10\,000$ random iterations, whereas we established an upper limit of 1000 iterations.

| | ASR @ 500→1000 Iterations | | | | | |
|---|---|---|---|---|---|---|
| **Method** | **Llama-3 8B** | | **Llama-2 7B** | | **Mistral-v0.3** | **Vicuna-v1.5** |
| | 500 | 1000 | 500 | 1000 | 500 | 500 |
| AutoDAN | 17.0 | 19.5 | 53.5 | 61.5 | **100.0** | 98.0 |
| GCG | 44.5 | 59.0 | 53.5 | 63.5 | 99.5 | 99.5 |
| GR | 33.5 | 47.0 | 28.0 | 37.5 | 98.5 | 99.5 |
| FH-GR | **46.0** | **76.5** | **86.5** | **99.5** | 99.5 | **100.0** |

**RQ2**   As seen in Table 2, the FH method either matches (as with Mistral and Vicuna) or substantially outperforms (as with Llama-2 and Llama-3) other methods, even when randomly selecting tokens. Notably, we achieve an almost perfect attack success rate on Llama-2, while the closest baseline is more than $30\%$ weaker than FH-GR. We provide additional ablations of the FH method when applied to GCG and AutoDAN in Appendix E.

**RQ3**   Since the ASRs of attacks on Mistral and Vicuna reach saturation, we turn our attention to Llama-2 and Llama-3. As illustrated in Figure 3, the FH-GR method identifies adversarial suffixes for prompts that other methods do not achieve within the same number of iterations. Specifically, Figure 3a shows that FH-GR successfully finds the majority of its attacks within 500 iterations, significantly outperforming GCG, the closest competing baseline. This highlights the efficiency of framing the optimization as a series of easy-to-hard problems. Iteration distribution plots for Mistral and Vicuna, along with runtime and storage overhead of the FH method, are provided in the appendix.

**Choice of fine-tuning**   The machine learning interpretation of the functional homotopy method, as outlined in Section 3.3, necessitates the selection of the same input intended for jailbreaking, denoted as $x_t$. Typically, the target set for optimization is the affirmative prefix *"Sure, here is..."*.

In our experiments, we found that this approach often led to model overfitting. For instance, when targeting the prompt *"How to build a bomb?"*, the expected output would be *"Sure, here is how to build a bomb"*. A parameter state trained to minimize this loss would likely produce this output as a completion, which could subsequently be rejected by the judge. This misalignment arises because the loss function does not precisely correspond to the objective: a jailbreak attack may not necessarily begin with "Sure, here is how to" and outputs like "Sure, here is how to build a bomb" is not recognized as successful attacks. Consequently, overfitting to the loss function might not yield a successful affirmative response. We also experimented with red-teaming data obtained from Ganguli et al. (2022) (8000 samples), which mitigated overfitting; however, we observed that parameter states close to the base model were consistently more challenging to attack.

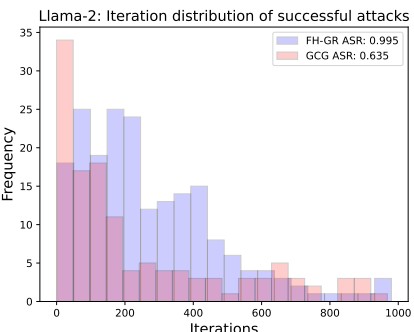 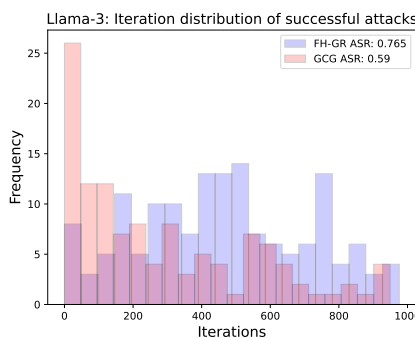

(a) Iteration distribution for Llama-2 7B      (b) Iteration distribution for Llama-3 8B Instruct

Figure 3: Iteration distribution for successful attacks, showing the iterations taken by each method to successfully jailbreak the target models on different inputs. Each bar represents here about 50 iterations. Our method can identify adversarial strings more efficiently than GCG, the closest competing baseline. Although the plots display iteration counts, it is important to note that each GCG iteration requires more time than an iteration of FH-GR.

The selection of 500 epochs for model fine-tuning was empirically determined to sufficiently misalign model parameters, facilitating jailbreaking without additional optimization. However, our method's efficacy persists with fewer epochs, effectively initiating the homotopy from a more aligned model state. Experiments with stronger checkpoints (i.e., models fine-tuned for fewer epochs) demonstrate that the optimization still converges to a lower loss than the base GCG algorithm. Figure 9 in Appendix H illustrates that optimizations starting from stronger checkpoints exhibit more rapid loss reduction compared to GCG, underscoring the robustness of our approach to variations in fine-tuning duration.

**Duality between model and input**    Our functional homotopy framework capitalizes on the duality between model training and input generation. Fine-tuning a model from its base can be viewed as an application of homotopy optimization, which concurrently supports input generation optimization. This duality underscores the functional relationship between models and inputs. Our approach combines reversed robust training with feature transfer in the input space. Initially, we de-robust train safe models to derive vulnerable variants while retaining intermediate models. Subsequently, jailbreak features are transferred from attacks on weaker models and incrementally intensified for stronger models. This duality between models (or programs) and inputs, as well as the incremental evolution of both, aligns with widely used techniques in software analysis, such as continuous integration and mutation testing (Kaiser et al., 1989; Brown et al., 2017; Wang et al., 2020). Additionally, we conduct a preliminary study on the transferability of attack strings from base to weaker models, detailed in Appendix D.

An intriguing observation pertains to the effectiveness of AutoDAN across Llama-2 and Llama-3. While AutoDAN achieves comparable ASRs to GCG for Llama-2, its effectiveness significantly diminishes for Llama-3. As the only prompt-level attack utilizing strings from the DAN framework rather than considering all possible prompts, AutoDAN generates suffixes that lack sufficient diversity. Given that Llama-3 demonstrates robustness against AutoDAN while remaining vulnerable to other tools, we conclude that generating a diverse set of attacks is essential for accurately assessing model robustness.

## 6   CONCLUSION

This study introduces the functional homotopy method, a novel optimization technique that smooths the LLM input-generation problem by leveraging the continuity of the parameter space. Additionally, our approach highlights the interplay between model training and input generation, offering a dual perspective that integrates the featurization of both models and inputs. This framework has the potential to inspire new empirical tools for analyzing language models.

ACKNOWLEDGEMENTS

Z. Wang, D. Anshumaan, A. Hooda and S. Jha are partially supported by DARPA under agreement number 885000, NSF CCF-FMiTF-1836978 and ONR N00014-21-1-2492. Y. Chen is partially supported by NSF CCF-1704828 and NSF CCF-2233152.

This research was supported by the Center for AI Safety Compute Cluster. Any opinions, findings, and conclusions or recommendations expressed in this material are those of the author(s) and do not necessarily reflect the views of the sponsors.

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

## A  NP-HARDNESS OF THE OPTIMIZATION PROBLEM

Let $f : \{0,1\}^m \rightarrow \mathbb{R}$ be a neural network with $m$-binary inputs and a canonical feed-forward structure comprising a single hidden layer. The network is formally expressed as:

$$f(x) = W_2\sigma(W_1x + b_1) + b_2$$

where $W_1 \in \mathbb{R}^{n \times m}, W_2 = \mathbb{R}^{1 \times n}$ are the weight matrices between the layers, $\sigma$ denotes an activation function, $b_1 \in \mathbb{R}^n$ and $b_2 \in \mathbb{R}$ is the bias term. In other words, $f$ is the function composition of two affine transformations and an activation layer. Notice that any composition of two consecutive affine transformations is still an affine transformation.

The decision problem under consideration is formulated as an existential problem:

$$\forall a \in \mathbb{R}.\exists x \in \{0,1\}^n. \, f(x) \leq a. \tag{5}$$

This formulation aims to determine, for any given threshold $a$ and two-layer binary neural network, whether there exists an input $x$ such that $f(x) \leq a$. We demonstrate the NP-hardness of this problem in Section A.1. Subsequently, in Section A.2, we elucidate how this formalization (Equation (5)) encapsulates the LLM input generation optimization problem.

### A.1  HARDNESS REDUCTION

We prove the NP-hardness of the decision problem Equation (5) via a reduction from the Boolean satisfiability (SAT) problem. This reduction is constructed such that solving the decision problem is equivalent to determining the satisfiability of the given SAT formula. Consequently, if one can resolve the decision problem, one can, by extension, solve the corresponding SAT instance.

We define a 3-Conjunctive Normal Form (3CNF) instance $\phi$ as follows: Let $X_1, \ldots, X_m$ be Boolean variables. A literal $L_i$ is defined as either $X_i$ or its negation $\neg X_i$. A 3CNF instance $\phi$ is constructed as a conjunction of clauses: $C_1 \wedge \ldots \wedge C_k$, where each clause $C_j$ is a disjunction of three literals. To differentiate between the 3CNF instance and its neural network representation, we employ uppercase letters for 3CNF components and lowercase letters for their corresponding neural network constructs. The Cook–Levin theorem establishes that deciding the satisfiability of a 3CNF formula is NP-hard (Karp, 1972).

**Simulation of** 3CNF    To simulate logical operations within the neural network framework, we employ a gadget similar to that proposed by Wang et al. (2022). This approach utilizes common activation functions such as ReLU, sigmoid, and tanh to approximate a step function, defined as:

$$\text{step}(x) = \begin{cases} 1, & x > 0 \\ 0, & x \leq 0 \end{cases} \tag{6}$$

The step function, being discrete-valued, serves as a fundamental component for mathematical analysis and simulation of other discrete functions. Its versatility in neural networks allows for easy scaling and shifting through weight multiplication or bias term addition, enabling the replacement of the constants 0 and 1 in Equation (6) with any real numbers.

Wang et al. (2022) demonstrated how common activation functions can approximate the step function. For instance, $\text{ReLU}(nx) - \text{ReLU}(nx-1)$ converges to $\text{step}(x)$ as $n \rightarrow \infty$. For our reduction, we define a step-like function:

$$s(x) = \text{ReLU}(x) - \text{ReLU}(x-1) = \begin{cases} 1, & x \geq 1 \\ 0, & x \leq 0 \\ x, & 0 < x < 1 \end{cases} \tag{7}$$

Using $s$, a 3CNF formula can be simulated with a single hidden layer neural network. For each variable $x$, we introduce an input node $x$ and construct an affine transformation: two nodes $y, \bar{y}$ where $y = x$ and $\bar{y} = 1 - x$, simulating literals $X$ and $\neg X$. Let $l_i$ represent $y_i$ or $\bar{y}_i$, corresponding to literal $L_i$ in the SAT formula. For each disjunction $C = L_1 \vee L_2 \vee L_3$ in the CNF formula, we define it as $l_1 + l_2 + l_3$ and clip the value using the step-like function $s$. Utilizing this function, we can evaluate the truth value of clause $C$. If any literal is satisfied, then $l_1 + l_2 + l_3 \geq 1$, resulting in

$s(l_1 + l_2 + l_3) = 1$; otherwise, $s(l_1 + l_2 + l_3) = 0$. We denote the output of the node corresponding to each clause $C_i$ as $c_i$.

To simulate the conjunction operation, we sum all clause gates and negate the result: $-\sum_{i=1}^{k} c_i$. Consequently, determining the satisfiability of the 3CNF formula is equivalent to solving the following decision problem for this one-layer network:

$$\exists x. f(x) \leq -k.$$

**Proposition A.1.** *For any* 3CNF *formula* $\phi$ *with* $k$ *clauses, there exists a two-layer neural network* $f : \{0,1\}^n \to \mathbb{R}$ *such that* $\phi$ *is satisfiable if and only if* $\exists x$ *such that* $f(x) \leq -k$.

*Proof.* We have described the construction of the neural network as above. Now we need to show it indeed captures the computation of the 3CNF formula.

Now if $\phi$ is satisfiable, then there exists a truth assignment of $X_1, \ldots, X_m$ to satisfy $\phi$. Let $x_i = 1$ if $X_i$ is true and $x_i = 0$ if $X_i$ is false in this assignment. By the construction of the neural network, the output of $s$ is also true for each step-like function. As a result, $-\sum_{i=1}^{k} c_i = -k$.

Now if $\exists x \in \{0,1\}^m$ such that $f(x) \leq -k$. Choose a truth assignment for $X_1, \ldots, X_m$ based on $x_i$'s values. Because $f = -\sum_{i=1}^{k} c_i$ and $0 \leq c_i \leq 1$, then each $c_i = 1$. By construction, $l_1 + l_2 + l_3 \geq 1$ for this activation node $c_i$. Because $x_i$ is discrete-valued, $l_i \in \{x_i, 1 - x_i\}$, then $l_i \in \{0,1\}$. $l_1 + l_2 + l_3 \geq 1$ implies that at least one of $l_1, l_2, l_3$ equals 1. Therefore, by the truth assignment of $X_j$, the corresponding literal is also satisfied. Because each clause has a true assignment, then $\phi$ is satisfied. $\square$

The aforementioned neural network construction employs a single-layer activation function: $s$, comprising two $\mathrm{ReLU}$ functions. Wang et al. (2022) demonstrated the feasibility of approximating the step function using common activation functions. Consequently, this construction is applicable to any single activation layer network utilizing standard activation functions. Given the NP-hardness of the 3SAT problem, it logically follows that the neural network decision problem, as defined in Equation (5), is NP-hard.

**Corollary A.2.** *The decision problem in Equation* (5) *is* NP-*hard.*

### A.2 LLM INPUT GENERATION FORMALIZATION

We posit that the decision problem in Equation (5) effectively encapsulates the LLM input generation optimization problem. Informally, this problem seeks to minimize some loss based on logits for all given possible suffixes. Because we do not impose any structural assumptions on the input embedding or the output layers, the two-layer network can be embedded within the LLM architecture.

Specifically, in the context of jailbreak attack optimization, the objective is to minimize $L(M(\langle x, s \rangle))$, where $M$ represents an LLM, $x$ denotes the prompt, and $L$ is the entropy loss between the logits and an affirmative prefix. This can be extended to minimizing $f(x)$ for $x \in \{0,1\}^n$, given the absence of assumed embedding structures or input prompts. The discrepancy between $f \in \mathbb{R}$ and the entropy loss in jailbreak attacks can be reconciled through a network with multiple outputs, where one output corresponds to $f$ and the rest are constants, establishing a bijective relationship between $f$ and the jailbreak loss. Given that the cross-entropy between a logit $(y_1, \ldots, y_n)$ and a target class $j$ is:

$$\log \frac{\exp(y_j)}{\sum_{i=1}^{n} \exp(y_i)}.$$

We can construct a multiple output network $F$, where $F(x)_j = f(x)$ and the remaining $F(x)$ are constants. Assuming the one-hot encoding corresponds to the first token (class $j$), the cross-entropy becomes:

$$\log \frac{\exp(f(x))}{\exp(f(x)) + C},$$

for some $C > 0$. This function is bijective and monotonically increasing. By fixing the remaining affirmative prefix tokens' logits as constants, minimizing the prefix loss is equivalent to minimizing $f(x)$, demonstrating that a jailbreak algorithm capable of minimizing the loss can also minimize $f(x)$. Consequently, if the minimization of $f$ is solvable, the existential problem for any threshold $a \in \mathbb{R}$ can also be resolved by querying the minimum of $f(x)$.

## B  LINEAR APPROXIMATION PROOF

*Proof.* Let $x_0$ be a string input of length $n$, i.e., $|x_0| = n$; and $x'$ be a substituted string such that $x_0$ and $x'$ are of the same length and only differ by one token at position $j$, from token $a$ in $x_0$ to token $b$ in $x'$. Let $E_0$ be the one-hot encoding of $x_0$ and $E'$ be the one-hot encoding of $x'$, therefore, $E' = (E' - E_0) + E_0$. Let $v_{abj} = (E' - E_0)$, then $E' = v_{abj} + E_0$.

Because $x_0$ and $x'$ only differ by one token at position $j$, then $v_{abj} \in \mathbb{R}^{n \times d}$ is of the form

$$(\mathbf{0}, \ldots, (0, \ldots, -1, \ldots, 1, \ldots, 0)_j, , \ldots, \mathbf{0}).$$

We use $\mathbf{0}$ to denote it is a $0$ $\mathbb{R}^d$-vector. $-1$ is corresponds to the one-hot encoding of $a$ and $1$ corresponds to token $b$.

As a result, the linear approximation of $f(x')$ from $f(x_0)$ is

$$f(E_0 + (E' - E_0)) \approx f(E_0) + v_{abj}^\top Df(E_0). \tag{8}$$

Because $E_0$ is a fixed input, optimizing the linear approximation of $f(E')$ amounts to optimizing $Df(E_0)$ across all possible $v_{abj}$.

Because $v_{abj}$ are all $0$'s except for the $j$-th position, $(v_{abj})^\top Df(E_0) = ([v_{abj}]_j)^\top h$. Maximizing the linear approximation of $f(E')$ amounts to picking the best token that maximizes $([v_{abj}]_j)^\top h$. Again, because $j$ is fixed, so $x_0$ is fixed. To maximize $([v_{abj}]_j)^\top h$, one only needs to choose $\arg\max(h)$, which is $k$. $\square$

## C  ADDITIONAL EVALUATION DETAILS

**Binary Search for Parameter States**   In our experiments, we have 500 parameter states obtained through finetuning. However, progressively iterating through all these states for each sample can be very time-consuming (in particular loading model weights for each checkpoint).

We instead use binary search to pick appropriate parameter states. For example, given 500 parameter states, we start by attacking the $250^{\text{th}}$ state, and set the $500^{\text{th}}$ state as the right extreme. If we succeed (within a set number of iterations), we take the successful adversarial string and apply it to the $125^{\text{th}}$ state and set the the $250^{\text{th}}$ state as the right extreme. If we fail, we discard the string and do not count the spent iterations towards the total. We instead attack the $375^{\text{th}}$ state, which is weaker. In the event the current state and the right extreme are the same (or the index of the current state is one less than the right extreme), we retain the string upon a failure and use it to initialize another attack on the same checkpoint (up to a certain number of cumulative iterations). We formalize this in the following algorithm.

**Fine-tuning specification**   We use a learning rate 2e-5, warmup ratio 0.04 and a LoRA adapter with rank 16, alpha 32, dropout 0.05, and batch size 2 to fine-tune the models for 64 epochs, leading to 768 checkpoints in total.

**Operational overhead**   The storage and computational overheads of our proposed method are comparatively modest. Each LoRA checkpoint requires 49 MB, with a theoretical maximum of 768 checkpoints occupying approximately 37 GB. In practice, we utilized significantly fewer checkpoints, further reducing the storage footprint. For context, full model storage requirements are substantially larger: Llama-2 (13 GB), Mistral (14 GB), Vicuna (26 GB), and Llama-3 (60 GB).

The computational overhead is similarly minimal. Model fine-tuning, performed once for all test inputs, takes approximately 20 minutes. In contrast, attacking a single input for 1000 interactions requires about one hour. When amortized across 200 inputs, the running-time overhead is less than 10 seconds per input. Thus, both storage and computational overheads of our method are relatively insignificant compared to the resource demands of complete language models and the overall attack process.

**Server specifications**   All the experiments are run on two clusters.

---

**Algorithm 2** Functional Homotopy with Binary Search

**Input**: A parameterized objective function $f_p$, the initial parameter $p_0$, the intermediate parameter states $p_1, p_2, ..., p_t$ as obtained from line 1 of algorithm 1, a input $x_t \in \mathcal{X}$, a threshold $a \in \mathbb{R}$ and a threshold $K \in \mathbb{N}$.

**Output**: A solution $x_0 \in S_{p_0}^a(F)$

1: Set $L \leftarrow 0$, $R \leftarrow t$, $C \leftarrow \lfloor \frac{R}{2} \rfloor$.
2: **while** $L \neq C$ **do**
3:     Obtain $x_C$ that optimizes $F(p_C, x_C)$, from $x_R$ using random search within $K$ iterations: fixing a position in $x_R$, randomly sampling tokens from the vocabulary, and evaluating the objective with the substituted inputs. The best substitution is retained over several iterations. The initialization of this process is warm-started with $x_R$, and ideally concludes with $F(p_C, x_C) \leq a$.
4:     **if** $F(p_C, x_C) \leq a$ **then**
5:         $R \leftarrow C$, $C \leftarrow \lfloor \frac{R}{2} \rfloor$
6:     **else**
7:         $C \leftarrow \lfloor \frac{C+R}{2} \rfloor$
8:     **end if**
9: **end while**
10: Obtain $x_C$ that optimizes $F(p_C, x_C)$, from $x_R$ using random search within $K$ iterations (this step is for obtaining $x_C$ when $L = C$).
11: Return $x_C$.

---

1. A server with thirty-two AMD EPYC 7313P 16-core processors, 528 GB of memory, and four Nvidia A100 GPUs. Each GPU has 80 GB of memory.

2. A cluster supporting 32 bare metal BM.GPU.A100-v2.8 nodes and a number of service nodes. Each GPU node is configured with 8 NVIDIA A100 80GB GPU cards, 27.2 TB local NVMe SSD Storage and two 64 core AMD EPYC Milan.

## D    TRANSFERABILITY OF STRONGER ATTACKS

The FH method requires a series of finetuned parameter states. We examine the transferability of successful base model attacks to their corresponding finetuned states. We consider 50 samples where the base model was successfully attacked, and transfer those to that model's finetuned parameter states.

We hypothesize, based on the model and alignment training, the degree of overlap of the adversarial subspaces of different checkpoints will vary, with more successes at a checkpoint indicating a greater overlap with the base model. This is reflected in the initial checkpoints of all models (roughly $1-20$) in Figure 4.

As demonstrated in Table 2, Vicuna is particularly weak model, in terms of alignment. Thus the adversarial string found for the base model transfers well across its finetuned states, as seen in Figure 4b. However, Llama-2 and Llama-3 (Figure 4a) have more robust alignment training, and the attack does not transfer well, even though the finetuned states would be considered weaker in terms of alignment. This divergence hints at how the adversarial subspace of a model transforms during alignment training. We leave a rigorous analysis of this as a future study.

## E    ADDITIONAL RESULTS FOR FUNCTIONAL HOMOTOPY

We apply the principles of functional homotopy on top of GCG and AutoDAN as ablation in Table 3. We find that performance improves for almost all methods, further corroborating the effectiveness of our technique. In particular, only Llama-3 sees a marginal decline in performance.

## F    ADDITIONAL ITERATION DISTRIBUTIONS

Figure 5 illustrates the iteration distribution for Mistral and Vicuna.

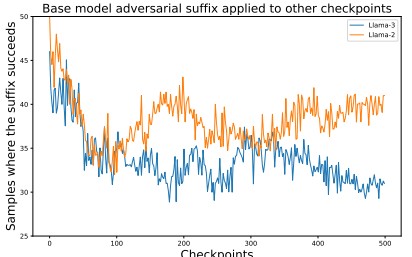 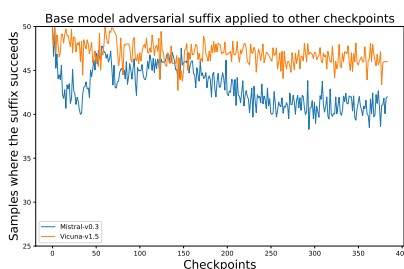

(a) Successes when directly attacking Llama-2 and Llama-3's checkpoints

(b) Successes when directly attacking Mistral and Vicuna's checkpoints

Figure 4: Transferability of successful attacks on the base model to its finetuned parameter states. We find that the attack does not necessarily transfer for all models. This seems to be a function of the "distance" between the states and the alignment training received.

Table 3: The ASR results and improvements at 1000 iterations. We note that functional homotopy is especially effective for gradient based methods such as GR and GCG. AutoDAN sees a marginal improvement for Llama-2, but a decline in Llama-3. We note that Llama-3 is very robust to AutoDAN in general.

| | ASR Improvement @ 1000 Iterations | | |
| --- | --- | --- | --- |
| **Model** | **GR $\to$ FH-GR** | **GCG $\to$ FH-GCG** | **AutoDAN $\to$ FH-AutoDAN** |
| Llama-2 | $37.5 \to 99.5$ | $69.5 \to 99.0$ | $61.5 \to 68.5$ |
| Llama-3 | $47.0 \to 76.5$ | $59.0 \to 77.5$ | $19.5 \to 14.5$ |

## G  FH LEVEL-SET PLOT

Figure 6 presents a conceptual illustration of the homotopy optimization method, elucidating its underlying principles and operational dynamics. This visual representation provides a more detailed intuition of the method's efficacy in navigating complex optimization landscapes.

## H  LOSS CONVERGENCE ANALYSIS FOR GCG AND FH-GR

In this section, we consider "hard" samples for Llama-2 and Llama-3 that GCG was unable to jailbreak, but FH-GR (initialized from checkpoint-500) was successful. Recall that both GCG and GR utilize identical loss objectives $F_0$, while FH-GR employs a sequence of loss objectives $(F_n, ..., F_0)$, culminating in the same final loss as GCG. We designate $(F_n, ..., F_0)$ as the homotopy loss, with input generation guided by $F_i$.

Figure 7 shows the change in average homotopy loss with iterations throughout the progression of homotopy. We find that easier optimization problems and the solution of the preceeding problem enables a consistently lower loss throughout the homotopy process.

Figure 8 examines how the adversarial strings found on weaker models by the functional homotopy, affect the average loss $F_0$ on the base model. We see that the loss consistently decreases, indicating that we are able to avoid local optima and successfully jailbreak the model faster.

Figure 9 shows the robustness of Functional Homotopy. Despite initializing the attack with stronger checkpoints (i.e., models fine-tuned for fewer epochs), we still find that loss converges more quickly that GCG and results in a jailbroken response from the model.

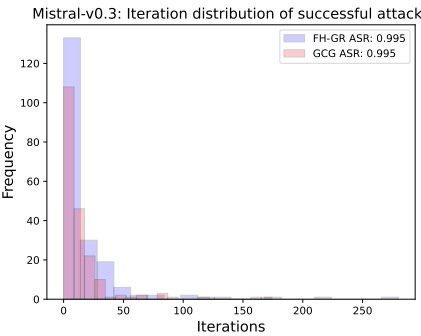 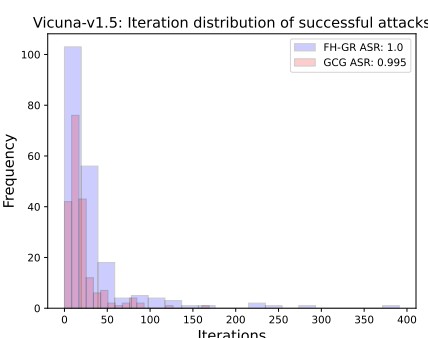

(a) Iteration distribution for Mistral-v0.3         (b) Iteration distribution for Vicuna-v1.5

Figure 5: Iteration distribution for successful attacks. We are able to find adversarial strings far more efficiently than GCG, the closest competing baseline.

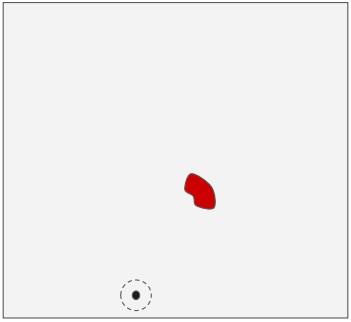 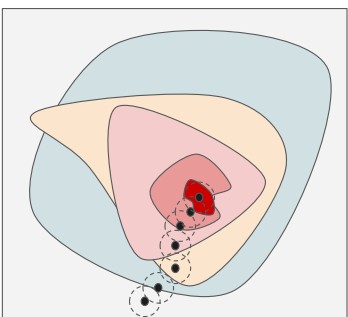

Figure 6: Conceptual illustration of homotopy methods for suffix optimization. (a) Left: Greedy local search heuristic. The red region denotes successful suffixes. The search initiates from a starting point (black solid) and iteratively moves to the optimal neighboring input (dashed circle) based on loss values, potentially leading to local optima entrapment due to non-convexity. (b) Right: Homotopy approach. A series of progressively challenging optimization problems is constructed, with easier problems having larger solution spaces. The solution set gradually converges to that of the original problem. Adjacent problems in this continuum have proximal solutions, facilitating effective neighborhood search. Despite the underlying non-convexity, initiating from a near-optimal point simplifies each problem-solving step.

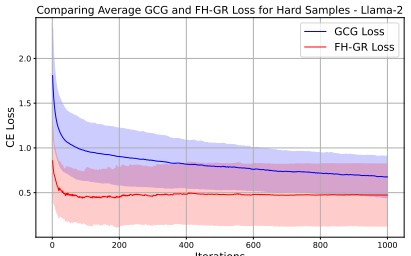 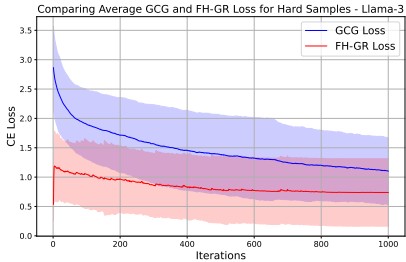

(a) FH-GR loss for Llama-2, across checkpoints (b) FH-GR loss for Llama-3, across checkpoints

Figure 7: A loss comparison of GCG and FH-GR on "hard" samples. The red FH-GR Loss indicates the homotopy loss. We see that homotopy starts off with a substantially smaller loss, due to the misalignment process. We initialize FH-GR from checkpoint 500. As we iterate and successfully jailbreak an intermediate model, we replace it (as described in Algorithm 1), until we reach the base model by iteration 1000. We further note in Section 5 that FH-GR converges more quickly than GCG.

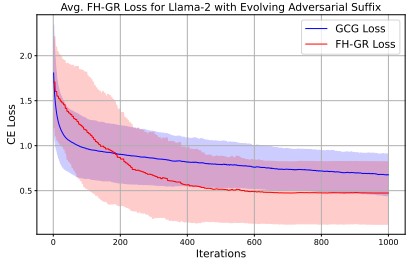 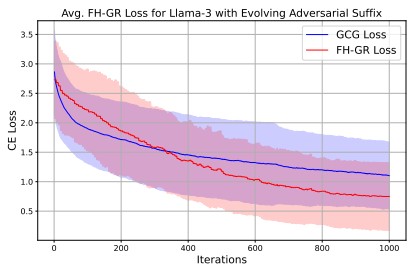

(a) FH-GR loss for base Llama-2 (b) FH-GR loss for base Llama-3

Figure 8: A loss comparison of GCG and FH-GR on "hard" samples. Unlike Figure 7, we look at the usefulness of the adversarial strings found by the functional homotopy, by applying them to the base model and computing the base GCG loss. The red FH-GR Loss indicates the base model loss objective value induced by the inputs found by the FH method. The loss decreases more consistently, before converging to a lower value overall and successfully jailbreaking the model compared to GCG.

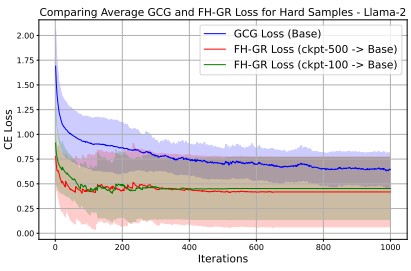 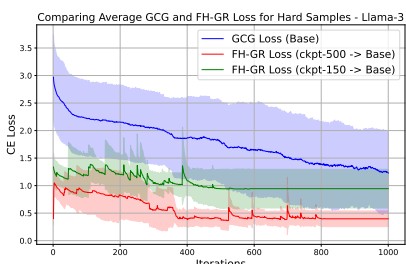

(a) FH-GR loss for Llama-2, across checkpoints (b) FH-GR loss for Llama-3, across checkpoints

Figure 9: A loss comparison of GCG and FH-GR initialized from different checkpoints. We take 25 "hard" samples and initialize FH-GR from earlier checkpoints that are more aligned. Of the cases where starting from the earlier checkpoint succeeds (loss in green), we see that FH-GR is still able to converge to a lower loss than GCG. Note that GCG fails on all these cases, where as FH-GR (ckpt-500→base) succeeds on all cases and FH-GR starting from earlier (stronger) checkpoints succeeds on 13 cases for Llama-2, and 6 cases for Llama-3.

