# OpenReview forum: "Functional Homotopy: Smoothing Discrete Optimization via Continuous Parameters for LLM Jailbreak Attacks"
_ICLR.cc/2025/Conference — ICLR 2025 Poster_

### Official Review · Reviewer_M9RT · 2024-10-28

**Soundness:** 2
**Presentation:** 3
**Contribution:** 3
**Rating:** 6
**Confidence:** 4

**Summary:**

This paper presents a novel approach called the Functional Homotopy (FH) method for solving discrete optimization problems in the context of jailbreak attacks on large language models (LLMs). The method leverages a duality between continuous model parameters and input generation, using an easy-to-hard problem-solving approach. The authors claim a 20%-30% improvement in success rates over existing methods when applied to LLMs such as Llama-2 and Llama-3.

**Strengths:**

* The paper introduces a novel functional homotopy method for jailbreak attacks on large language models (LLMs), building on the traditional homotopy framework used in optimization. Instead of directly solving the complex discrete optimization problem, the approach breaks it down into a series of progressively harder problems. Each solution from an easier problem is used as the starting point for the next, harder problem, which makes the optimization process smoother and more manageable.

* The experimental results show that this method delivers a 20%-30% improvement in success rates compared to baseline techniques when applied to state-of-the-art open-source models.

**Weaknesses:**

* The proposed method appears overly simplistic, lacking both theoretical guarantees and in-depth analysis. The randomized search approach is basic and does not provide any assurance of optimality. Additionally, the complexity of performing a successful randomized search seems exponential with respect to the dimensionality of the problem. A formal theoretical analysis offering guarantees for the method would significantly strengthen the contribution.
* The baselines used for comparison also seem relatively weak. For instance, the gradient-based methods are naturally expected to perform poorly, given their myopic nature and disconnection from true gradient information. Furthermore, the datasets used for the experiments are small, making it difficult to draw definitive conclusions about the method’s effectiveness and efficiency. Stronger baselines would provide a more meaningful comparison.
* The algorithm's performance seems to be highly dependent on hyperparameter choices, such as the threshold, number of random search steps, and the search radius. A more comprehensive analysis on how to select these hyperparameters would provide valuable insight and improve the robustness of the proposed approach.

**Questions:**

see weakness

---

> ### Author Response · Authors · 2024-11-24
>
> We appreciate your professional review and address your concerns here:
> ### Algorithmic Intuition, Effectiveness and Analysis
> We discuss the algorithmic intuition underlying our approach in the ["Algorithmic Intuition" comment](https://openreview.net/forum?id=uhaLuZcCjH&noteId=GTy6u4hr3W). Our study is primarily an empirical methodology, drawing inspiration from homotopy optimization techniques.
>
> Fundamentally, most token-wise optimization algorithms utilize greedy search heuristics, which form the foundation of state-of-the-art methods in this domain [1,2,5,6,9,10]. However, greedy search is susceptible to **local optima entrapment** in non-convex optimization landscapes.
>
> Our FH method effectively mitigates this limitation. To substantiate our claims, we have provided additional empirical analyses showing the effectiveness of our approach in the optimization process. These findings are detailed in the ["Algorithmic Intuition" comment](https://openreview.net/forum?id=uhaLuZcCjH&noteId=GTy6u4hr3W) and illustrated in **Figures 7 and 8 (Appendix F)** of the revised manuscript.
>
> The theoretical analysis of optimality guarantees for our FH method presents significant challenges and extends beyond the scope of this paper, particularly considering that the underlying homotopy is constructed via model training, which is inherently difficult to analyze. Even for simpler architectures like feedforward neural networks, the underlying optimization problems are **typically NP-hard**, and optimization algorithms rarely offer optimality guarantees [7, 8].
>
> ---
> ### Baselines and Datasets
> Our evaluation employs widely recognized baselines from literature ([1, 2]). We focus on optimization-based attacks, excluding black-box methods, as our technique is designed to augment existing optimization approaches. We welcome suggestions for additional relevant baselines.
>
> The datasets used in our benchmarks contain approximately 500 samples (AdvBench [1] and HarmBench [3]). Given that individual runtimes for each method can exceed one hour per instance on our GPUs, we selected a sample size of 200 instances to balance performance evaluation with practical time and resource constraints. This sample size is consistent with or exceeds that of similar jailbreak studies, which used 50 and 183 examples in their evaluations [6,10].
>
> ---
> ### Hyperparameters and Robustness
> We discuss the specific overhead requirements for our method, including storage and fine-tuning costs, in the ["Hyperparameters" comment](https://openreview.net/forum?id=uhaLuZcCjH&noteId=dX3VWD8DH6). All other parameters mentioned, have been adopted from existing literature for a fair comparison:
> 1. The loss threshold for our method is determined by the LLM judge's decision on jailbreak success, as described in line 404.
> 2. The number of random search steps is derived from the baseline methods GCG [1] and AutoDAN [2].
> 3. The search radius (20 tokens) is adopted from [1] which is also utilized in several other comparative studies (such as [5],[6]).
>
> ---
> We welcome any additional questions you may have and are happy to address them. If you believe that our responses and the revisions to the manuscript have adequately addressed your concerns and enhanced its quality, we kindly request that you consider increasing the score for our paper.
>
> ---
> ### References
> [1] Zou et al. “Universal and Transferable Adversarial Attacks on Aligned Language Models” arXiv preprint arXiv:2307.15043 (2023)
>
> [2] Xiaogeng Liu, Nan Xu, Muhao Chen, and Chaowei Xiao. “AutoDAN: Generating stealthy jailbreak prompts on aligned large language models. In The Twelfth International Conference on Learning Representations”, 2024a. URL https://openreview.net/forum?id=7Jwpw4qKkb.
>
> [3] Mantas Mazeika, Long Phan, Xuwang Yin, Andy Zou, Zifan Wang, Norman Mu, Elham Sakhaee, Nathaniel Li, Steven Basart, Bo Li, et al. “Harmbench: A standardized evaluation framework for automated red teaming and robust refusal.” arXiv preprint arXiv:2402.04249, 2024.
>
> [5] Hayase, Jonathan, et al. "Query-based adversarial prompt generation." arXiv preprint arXiv:2402.12329 (2024).
>
> [6] Andriushchenko, Maksym, Francesco Croce, and Nicolas Flammarion. "Jailbreaking leading safety-aligned llms with simple adaptive attacks." arXiv preprint arXiv:2404.02151 (2024)
>
> [7] Vincent Froese, Christoph Hertrich. “Training Neural Networks is NP-Hard in Fixed Dimension”. In The Thirty-seventh Conference on Neural Information Processing Systems. (2023)
>
> [8] G. Katz, C. Barrett, D. Dill, K. Julian and M. Kochenderfer. “Reluplex: An Efficient SMT Solver for Verifying Deep Neural Networks.” Proc. 29th Int. Conf. on Computer Aided Verification (CAV). Heidelberg, Germany, July 2017.
>
> [9] Sadasivan, Vinu Sankar, et al. "Fast Adversarial Attacks on Language Models In One GPU Minute." arXiv preprint arXiv:2402.15570 (2024).
>
> [10] Mehrotra, Anay, et al. "Tree of attacks: Jailbreaking black-box llms automatically." arXiv preprint arXiv:2312.02119 (2023).

---

> ### Comment · Reviewer_M9RT · 2024-11-26
> **answer**
>
> I appreciate the authors' rebuttal and the inclusion of additional experiments and increased my score. However, I believe some of my concerns remain unresolved. For instance, while the authors argue that greedy search is prone to local optima entrapment, the proposed FH approach lacks guarantees or comprehensive analysis demonstrating its ability to escape local minima effectively. Additionally, while the baseline is widely recognized, is it truly representative of the state-of-the-art in the literature?

---

> ### Author Response · Authors · 2024-12-02
>
> We sincerely appreciate your thoughtful review and the decision to raise the score. We are grateful for your additional insights and would like to address them respectfully:
>
> ---
> ## Guarantees or comprehensive analysis
> The present study does not provide theoretical guarantees for the proposed algorithms, a limitation shared by existing jailbreak attack methodologies to the authors' knowledge. However, we provided a formal proof of the NP-hardness of the jailbreak attack optimization problem. This result underscores the inherent difficulty in establishing theoretical guarantees and time complexity bounds for such algorithms. Consequently, the research emphasis has been placed on empirical validation of the proposed methods' efficacy.
>
> ### Hardness result
> We have formally demonstrated the NP-hardness of the jailbreak attack optimization problem, as elaborated in the ["NP-Hardness of Jailbreak Attacks" comment](https://openreview.net/forum?id=uhaLuZcCjH&noteId=rm78wqFf8I). This complexity result suggests that efficient algorithms for LLM jailbreak optimization likely require the exploitation of problem-specific structures. The proposed FH method, which leverages model fine-tuning, can be interpreted as utilizing the "alignment" property of the models. A rigorous analysis of the FH approach would necessitate an exploration of neural network structures, LLM architectures, and typical language data characteristics, pushing the boundaries of current deep learning and LLM theory. While providing theoretical analysis and guarantees presents an intriguing avenue for future research, it exceeds the scope of the current study and the relevant literature.
>
> ### Preliminary empirical analysis
> We have included a preliminary empirical analysis in the ["Algorithmic Intuition" comment](https://openreview.net/forum?id=uhaLuZcCjH&noteId=GTy6u4hr3W), which supports our method's assumptions and effectiveness.
> Our findings regarding the utility of intermediate model checkpoints in facilitating attacks on the base model have significant implications for the deep learning community and may inspire future work.
>
> ---
> ## State-of-the-art baselines
> For state-of-the-art baselines, we refer to the recent ICML benchmark paper [1], where Figure 5 indicates AutoDAN and GCG as the most effective attacks.
>
> ---
> We sincerely appreciate your valuable feedback and hope that clarifications address your concerns. We remain open to any further suggestions that could enhance our work. If you find that our responses and new results have sufficiently addressed your concerns and improved its quality, we would be grateful if you would consider increasing the score for our paper. Thank you for your consideration.
>
> ---
> ## References
> [1] Mantas Mazeika, Long Phan, Xuwang Yin, Andy Zou, Zifan Wang, Norman Mu, Elham Sakhaee, Nathaniel Li, Steven Basart, Bo Li, et al. “Harmbench: A standardized evaluation framework for automated red teaming and robust refusal.” ICML 2024.

---

> > ### Comment · Reviewer_M9RT · 2024-12-02
> > **answer**
> >
> > I'd like to thank the authors again for their effort and engagement with my concerns! Please include all these new analysis in the revision of the paper and I have raised my score again.

---

> > > ### Author Response · Authors · 2024-12-02
> > >
> > > We want to express our sincere gratitude for your decision to increase the score once again. We will update the camera-ready version of the paper accordingly.

---

### Official Review · Reviewer_bVKA · 2024-11-01

**Soundness:** 2
**Presentation:** 1
**Contribution:** 3
**Rating:** 8
**Confidence:** 3

**Summary:**

The paper trains the LLM to be less aligned and then attacks it with a greedy search for token replacements on the weaker versions of the model, gradually increasing the difficulty by reverting step by step to the aligned version.

**Strengths:**

1. The idea is interesting and novel in its approach to manipulating the model's parameters.
2. The empirical performance in terms of ASR appears strong.
3. The effort to provide evidence that the token gradient in GCG is not beneficial is appreciated, even though this has already been hypothesized.

**Weaknesses:**

1. The number of steps used for 'unalignment' must be decided beforehand. The procedure, if I understand correctly, is: conduct *N* steps to transition from parameter *p* to unaligned parameters *p'*. Then, iterate back and find an attack suffix for each *p_i* based on the previous one. This makes it difficult to determine if *p'* is sufficiently unaligned, and during the process of creating a new suffix, it is not possible to adjust to easier parameters when the attack does not seem effective.
2. The method introduces many additional hyperparameters (e.g., number of fine-tuning steps, number of GR steps per parameter set) and operational overhead, which are not discussed adequately (not even in RQ3). The authors should compare the attacks based on computational cost rather than steps, as fine-tuning is not factored into their 'step' metric.
3. The writing needs improvement. The text is often verbose, repeating the same points at different locations, and could be more concise. Additionally, there are inconsistencies in mathematical symbols, such as *p* being defined as the input (prompt) but is later used to refer to model parameters.

**Questions:**

1. I am curious about why the method works. I would understand if gradient optimization were involved and it was argued that the landscape is smoother, guiding the process to the next optimum. However, since the suffix is optimized with a greedy search, why is this more effective? If the main idea is just remaining at a broad minimum during random search, then the transferability from the base model to the weakest model should be higher, which does not appear to be the case.
2. In Figure 3, what does it mean for FH-GR to be a successful attack within 1 iteration? Does this mean only one parameter update, or does it start from the weakest model after *N* updates and do only one GR search?

### Review Summary:
The paper presents an interesting idea. However, in its current state, I am not in favor of accepting it. I believe more research into the method is necessary, further evidence should be gathered, and the writing should be improved. That said, I am open to revising my score if my concerns are addressed.

---

> ### Author Response · Authors · 2024-11-24
>
> We appreciate your professional review and address your concerns here:
>
> ### Number of Unalignment Steps
> We 'unalign' the model parameters until jailbreaking occurs without input optimization, which is achieved after 500 epochs of fine-tuning for all evaluated models. However, **this extensive misalignment is not essential**. The FH method remains effective with fewer epochs, initiating from a more aligned model state.
>
> Additional experiments with stronger checkpoints (i.e., models fine-tuned for fewer epochs) demonstrate that optimization still converges faster to a lower loss compared to the base GCG algorithm. Detailed results are presented in the ["Hyperparameters" comment](https://openreview.net/forum?id=uhaLuZcCjH&noteId=dX3VWD8DH6) under "Robustness of Hyperparameter Tuning".
>
> ---
> ### Hyperparameters and Computation Overheads
> Our method incurs minimal runtime overhead when amortized across multiple inputs. The one-time fine-tuning process for each model takes approximately 20 minutes, while attacking a single input for 1000 iterations requires at least 45 minutes. When distributed across 200 inputs, the additional runtime per instance is **approximately 6 seconds**. Detailed individual runtimes are presented in the ["Hyperparameters" comment](https://openreview.net/forum?id=uhaLuZcCjH&noteId=dX3VWD8DH6).
>
> Notably, even with this additional time, FH-GR often demonstrates lower overall runtime compared to baselines due to its higher ASR and earlier termination in many cases.
>
> ---
> ### Notation Inconsistencies and Conciseness
> We have revised the paper to make the paper more concise: we removed repeated points that appear in the introduction and main method sections. We also fixed the symbol inconsistencies of the paper.
>
> ---
> ### Algorithmic Intuituion
> Greedy search algorithms, including gradient descent, are generally effective for convex optimization problems due to the equivalence of local and global optima. However, they are susceptible to local optimum entrapment in non-convex scenarios. Our homotopy approach constructs a series of effective intermediate solutions converging to the original problem's solutions. We illustrate this concept through a new level-set visualization in **Figure 6, Appendix E**, with detailed explanations provided in the ["Algorithmic Intuition" comment](https://openreview.net/forum?id=uhaLuZcCjH&noteId=GTy6u4hr3W).
>
> Furthermore, we present additional empirical analysis demonstrating the occurrence of local optimum entrapment and how our functional homotopy method mitigates this issue, as measured by the original optimization objective. This analysis is depicted in **Figures 7 and 8 in Appendix F**, with comprehensive explanations available in the ["Algorithmic Intuition" comment](https://openreview.net/forum?id=uhaLuZcCjH&noteId=GTy6u4hr3W).
>
> ---
> ### Successful Attacks by FH-GR
> To clarify, each bar in figure 3 represents 50 iterations of the attack. If a method succeeds in that first bar, it indicates that the base model was successfully broken within 50 iterations. We also updated the plot caption in the revised manuscript to avoid this confusion.
>
> ---
> We welcome any additional questions you may have and are happy to address them. If you believe that our responses and the revisions to the manuscript have adequately addressed your concerns and enhanced its quality, we kindly request that you consider increasing the score for our paper.

---

> > ### Comment · Reviewer_bVKA · 2024-11-25
> >
> > I appreciate the extensive rebuttal, the revisions of the paper and the additional experiments. I would like to see some of the new experiments in the main text of the paper for the camera version, but I am happy with the answers and have increased my score.

---

> > > ### Author Response · Authors · 2024-11-25
> > >
> > > We sincerely appreciate your decision to increase the score and will update the camera-ready version of the paper accordingly.

---

### Official Review · Reviewer_vykH · 2024-11-02

**Soundness:** 3
**Presentation:** 3
**Contribution:** 3
**Rating:** 6
**Confidence:** 2

**Summary:**

This paper introduces the Functional Homotopy (FH) Method, a novel optimization approach designed to tackle the discrete optimization challenges inherent in language models, with a specific application to synthesizing jailbreak attacks. The authors note that existing gradient-based techniques, effective in continuous spaces like image models, face difficulties when applied to the discrete token space of language models. The FH method addresses this by leveraging a functional duality between model training and input generation. The method involves transforming a difficult optimization problem into a series of easier ones, using a homotopy-based approach where model parameters and inputs are optimized in tandem. The paper demonstrates that this method achieves a significant improvement in attack success rates (20-30%) over existing techniques when tested on open-source models like Llama-2 and Llama-3.

**Strengths:**

The paper introduces an original optimization method, blending continuous and discrete optimization techniques in a way that has not been widely explored for language models. This approach brings a fresh perspective to the field of adversarial attacks.  The authors provide a strong theoretical foundation for their method, clearly explaining the duality between model training and input generation. The homotopy-based approach is well-grounded in optimization literature, which adds depth to the contribution. The demonstrated success of the FH method in improving attack success rates against robust language models is significant. The practical implications for AI safety research are noteworthy, especially in understanding and defending against potential vulnerabilities in LLMs. The paper is clearly written, with a logical structure that effectively guides readers through the problem, methodology, and experimental results. The use of diagrams, such as the illustration of the homotopy path, aids in understanding the method’s workings.

**Weaknesses:**

The paper does not thoroughly address the computational demands of the FH method. Randomly sample tokens
from the vocabulary to replace the token at that position can be time-consuming. Given the potential resource constraints in real-world applications, more details on runtime efficiency and how the method scales with model size would be beneficial.

**Questions:**

The authors mention issues with overfitting to specific malicious queries, especially when targeting outputs that models are trained to reject. Are there additional strategies or techniques that could be implemented to reduce this risk without compromising attack success rates?

While the method shows improvements in success rates, could the authors provide more comprehensive comparisons of computational efficiency, such as runtime and resource usage, relative to baseline methods? This would help in understanding the practical trade-offs of deploying the FH method.

---

> ### Author Response · Authors · 2024-11-24
>
> We appreciate your positive review and address your specific concerns as follows:
>
> 1. We would like to clarify that each method runs for **at most 1000 iterations** for a given sample, and the individual runtimes for each method (including the check for a successful jailbreak, which happens for all methods) can be found in the ["Hyperparameters" comment](https://openreview.net/forum?id=uhaLuZcCjH&noteId=dX3VWD8DH6).
>
> As we can see, FH-GR in fact takes the least amount of time to run. An increase in model size will increase the time taken to calculate gradients, as well as the time taken for a forward pass.
> * Increased times to calculate gradients will increase the time to run GCG.
> * Increased times to calculate a forward pass will affect AutoDAN more than GR or GCG, since AutoDAN generates adversarial inputs with 90 tokens on average.
>
> In such a case, GR will have the least increase in runtimes.
>
> 2. One method to achieve this is to use jailbroken responses from another model as the target string, instead of *“Sure, here is…”*.
>
> 3. We present comprehensive comparisons of computational efficiency in the general response ["Hyperparameters" comment](https://openreview.net/forum?id=uhaLuZcCjH&noteId=dX3VWD8DH6). Overall, our proposed method demonstrates comparatively modest storage and computational overheads.
>
> We welcome any additional questions you may have and are happy to address them. If you believe that our responses and the revisions to the manuscript have adequately addressed your concerns and enhanced its quality, we kindly request that you consider increasing the score for our paper.

---

### Official Review · Reviewer_BDPK · 2024-11-04

**Soundness:** 3
**Presentation:** 3
**Contribution:** 3
**Rating:** 8
**Confidence:** 2

**Summary:**

The paper proposes the functional homotopy method, a novel optimization approach for LLM attacks. FH fintunes the LLM and solves a number of easy-to-hard problems, which are constructed via gradient descent. The results show a substantial improvement over different baselines.

**Strengths:**

- The paper is well-written and easy to follow
- The methodology is novel and well-motivated
- The proposed attack allows easy substitution of future attacks
- FH shows promising results

**Weaknesses:**

- L81-82: "can retain" should be "have to retain"
- The efficiency comparison is based on the iterations instead of the runtime, which seems to be a fairer comparison.
- Missing related work. Consider including [1, 2]


[1] Schwinn, Leo, David Dobre, Sophie Xhonneux, Gauthier Gidel, and Stephan Gunnemann. "Soft prompt threats: Attacking safety alignment and unlearning in open-source llms through the embedding space." arXiv preprint arXiv:2402.09063 (2024).
[2] Liao, Zeyi, and Huan Sun. "Amplegcg: Learning a universal and transferable generative model of adversarial suffixes for jailbreaking both open and closed llms." arXiv preprint arXiv:2404.07921 (2024).

**Questions:**

Questions:
- How much storage does it take to store all the LLMs?
- Is there ablation using another baseline for the discrete optimization for FH, e.g., AutoDAN or GCG?

---

> ### Author Response · Authors · 2024-11-24
>
> We appreciate your thorough review and address your specific concerns as follows:
>
> 1. Lines 81-82: We have revised the manuscript as suggested.
> 2. Runtime Information: We have included runtime information, including overhead details, in the ["Hyperparameters" comment](https://openreview.net/forum?id=uhaLuZcCjH&noteId=dX3VWD8DH6). Notably, our homotopy method's runtime overhead is minimal compared to the computational cost of running the models for hundreds of iterations during the attack process.
> 3. We thank you for sharing these related works, which have been incorporated into the revised manuscript.
> 4. Storage Overhead: We have detailed the storage requirements in the ["Hyperparameters" comment](https://openreview.net/forum?id=uhaLuZcCjH&noteId=dX3VWD8DH6). Specifically, our method employs LoRA for fine-tuning, with each checkpoint requiring **49 MB**. The maximum theoretical storage requirement is approximately 38 GB for 768 checkpoints, although in practice, we utilized significantly fewer, thereby reducing actual storage needs. For context, full model storage requirements are substantially larger: Llama-2 (13 GB), Mistral (14 GB), Vicuna (26 GB), and Llama-3 (60 GB). Thus, our method's storage overhead is comparatively modest relative to full model storage requirements.
> 5. We have not yet completed the benchmarking for the ablation study; however, we will promptly inform you of the results once the experiments are concluded.
>
> We welcome any additional questions you may have and are happy to address them. If you believe that our responses and the revisions to the manuscript have adequately addressed your concerns and enhanced its quality, we kindly request that you consider increasing the score for our paper.

---

> > ### Comment · Reviewer_BDPK · 2024-11-27
> >
> > I thank the authors for their response.
> >
> > My concerns have been addressed and I raise my score. I think the paper proposes an interesting idea and is a valuable contribution to LLM attacks.
> >
> > If the time allows, I suggest the authors to include point 5 in the final version as I think it strengthens the paper.

---

> ### Author Response · Authors · 2024-11-27
>
> We extend our sincere gratitude for your score adjustment. We will include the experimental results for point 5 in our camera-ready version.
>
> For your interest, we have obtained preliminary results (Attack Success Rates) for point 5 on **FH-GCG** using a microbenchmark of 50 samples. All attacks are run for 1000 iterations. These initial findings further corroborate the effectiveness of our method:
>
> |  Model | FH-GCG  | GCG  |
> |---|---|---|
> |  Llama-2 | 98.0  | 58.0  |
> |  Llama-3 | 84.0  |  56.0 |
> |  Mistral |  100.0 |  100.0 |
> |  Vicuna |  100.0 |  100.0 |
>
> Please let us know if you have any additional questions or concerns. We are more than happy to address them.

---

> > ### Comment · Reviewer_BDPK · 2024-12-02
> >
> > I thank the authors for sharing the results. I maintain my positive score.

---

### Author Response · Authors · 2024-11-24
**General Response**

We express our gratitude to all reviewers for their thorough evaluation and insightful inquiries, which have led to further analysis and significant improvements in our paper. We particularly appreciate the unanimous recognition of our method's **originality**, **novelty**, and **empirical superiority**. Our study provides compelling evidence that intermediate model checkpoints can facilitate attacks on the base model, a finding with broad implications for the deep learning community, given that model fine-tuning LLM is a standard practice known to potentially compromise model alignment [1].

We address several common concerns raised during the review process in our other comments **("Algorithmic Intuition" and "Hyperparameters")**.

All addressed concerns have been incorporated into the revised manuscript, with new text highlighted in blue.

---
## References
[1] Xiangyu Qi, Yi Zeng, Tinghao Xie, Pin-Yu Chen, Ruoxi Jia, Prateek Mittal, and Peter Henderson. "Fine-tuning Aligned Language Models Compromises Safety, Even When Users Do Not Intend To!." . In The Twelfth International Conference on Learning Representations.2024.

---

### Author Response · Authors · 2024-11-24
**Algorithmic Intuition**

We emphasize two key aspects that differentiate our approach from a purely random search: (1) the discrete topology defined by Hamming distance, and (2) the utilization of greedy search heuristics in optimization algorithms, including GCG and GR.
1. The discrete topology is characterized by a Hamming distance of 1 between neighboring inputs that differ by a single token. Given our focus on 20-token suffixes, the maximum distance between any two suffixes is 20. Theoretically, this implies that an existing attack suffix could be identified within 20 iterations under ideal conditions.
2. Our optimization algorithms employ local search with greedy heuristics. Each iteration samples a fixed number of neighboring inputs at distance 1 from a seed input, selecting the neighbor with the minimal objective value as the seed for subsequent iterations. However, this greedy approach may lead to entrapment in local optima, as proximal neighbors to a successful suffix do not necessarily yield lower loss values.

While both GCG and GR adhere to this framework, GCG incorporates gradient information for neighbor sampling, offering only marginal improvements over random sampling.

To address the limitations of greedy search heuristics, we propose a **homotopy-based approach**. This method constructs a series of progressively complex optimization problems through continuous model fine-tuning. The underlying principle posits that solutions to adjacent problems within this continuum are likely proximal, facilitating more effective search guidance towards the global optimum compared to local greedy heuristics. Our FH method generates a series of effective intermediate solutions converging to the original problem's solutions.  We provide a new level-set based conceptual illustration **(Figure 6 in Appendix E)** to show the evolution of the solution underpinning our homotopy method.

We present additional empirical analyses to substantiate this argument. Recall that both GCG and GR utilize identical loss objectives (F_0), while FH-GR employs a sequence of loss objectives (F_n, ..., F_0), culminating in the same final loss expression as GCG. We designate (F_n, ..., F_0) as the homotopy loss, with inputs guided by F_i.

The evolution of the loss objective is graphically represented over 1000 iterations of input search. For FH-GR, we present two plots: (1) the progression of homotopy loss **(Figure 7 in Appendix F)**, and (2) the evolution of F_0 induced by the inputs during FH-GR optimization **(Figure 8 in Appendix F)**.

---
## Empirical Results
The empirical evidence aligns with our homotopy method assumptions: (1) The progression of homotopy loss demonstrates that introducing easier problems enables the maintenance of lower homotopy loss during optimization. (2) Comparing F_0 evolution for GCG and FH-GR inputs reveals that GCG inputs, driven by greedy search heuristics, rapidly decrease loss initially but soon stagnate, likely due to local optima entrapment. Conversely, FH-GR inputs, guided by homotopy objectives, exhibit a slow but steady decrease in loss, ultimately achieving a **significantly lower loss**.

---

### Author Response · Authors · 2024-11-24
**Hyperparameters**

### Computational Overhead
Our proposed method demonstrates comparatively modest storage and computational overheads. We use LoRA [1], which is a parameter-efficient method for finetuning. **Each LoRA checkpoint requires 49 MB**, with a maximum of 768 checkpoints occupying approximately 38 GB. In practice, we utilized significantly fewer checkpoints, further reducing storage requirements. For context, full model storage needs are substantially larger: Llama-2 (13 GB), Mistral (14 GB), Vicuna (26 GB), and Llama-3 (60 GB)

We also provide the average runtimes (in minutes) of different methods (including the check for success performed by the judge) for a single test case while ensuring maximal GPU utilization.

| Model |  FH-GR |  GCG |  AutoDAN |
|---|---|---|---|
|  Llama-2 (1000 iters)| 49.45  |  50.01 |  120.08 |
|  Llama-3 (1000 iters)| 46.08  |  46.08 |  81.34 |
|  Mistral (500 iters)| 23.35  |  23.82 |  67.53 |
|  Vicuna (500 iters)| 24.09  |   24.47 |  63.16 |


Recall that we only fine-tuned each model once for all test inputs, which takes **approximately 20 minutes**, while attacking a single input for 1000 iterations requires **at least 45 minutes**. When amortized across 200 inputs, the running-time overhead is about **6 seconds per instance**. Consequently, both the storage and computational overheads of our method are relatively insignificant compared to the resource demands of complete language models and the overall attack process. We have included detailed overhead information in **Appendix A** of the updated manuscript.

---
### Robustness of Hyperparameter Tuning
 The selection of 500 epochs for model fine-tuning was based on empirical observations that this duration sufficiently misaligns the model parameters to facilitate jailbreaking **without additional optimization**. However, our method's effectiveness persists with fewer epochs, essentially initiating the homotopy from a more aligned model state.

We conducted additional experiments with stronger checkpoints (i.e., models fine-tuned for fewer epochs), and the result demonstrates that the optimization still converges to a lower loss than the base GCG algorithm. **Figure 9 in Appendix F** illustrates that optimizations starting from stronger checkpoints exhibit more rapid loss reduction compared to GCG, underscoring the robustness of our approach to variations in the fine-tuning duration.

---
### Search Steps and Radius
1. We use 1000 search steps, as per the baseline methods GCG [2] and AutoDAN [3] in order to make a faithful comparison with them.
2. We consider a search radius of 20 tokens for the adversarial suffix, as per GCG [2], in order to make a faithful comparison with it and AutoDAN, which also uses the same baseline in it's comparisons.

---
### References
[1] Edward J Hu, Yelong Shen, Phillip Wallis, Zeyuan Allen-Zhu, Yuanzhi Li, Shean Wang, Lu Wang, & Weizhu Chen (2022). LoRA: Low-Rank Adaptation of Large Language Models. In International Conference on Learning Representations.

[2] Zou et al. “Universal and Transferable Adversarial Attacks on Aligned Language Models” arXiv preprint arXiv:2307.15043 (2023)

[3] Xiaogeng Liu, Nan Xu, Muhao Chen, and Chaowei Xiao. “AutoDAN: Generating stealthy jailbreak prompts on aligned large language models. In The Twelfth International Conference on Learning Representations”, 2024a. URL https://openreview.net/forum?id=7Jwpw4qKkb.

---

### Author Response · Authors · 2024-12-02
**NP-Hardness of Jailbreak Attacks**

We additionally established that the computational complexity of the jailbreak attack optimization problem is **NP-hard**. This problem involves minimizing the prefix-matching loss (i.e. cross-entropy loss with the target string *"Sure here is..."*), for a given prompt in any LLM.

This is proven through a demonstration that a **single hidden layer** neural network with binary inputs can emulate the computation of a boolean formula. Due to revision constraints, the detailed proof is available [anonymously at this link](https://anonymous.4open.science/r/np-hard-52F1/NP_hardness.pdf), and will be incorporated into the camera-ready version.

This study establishes the computational complexity of the LLM jailbreak optimization problem, a new contribution to the field as previous jailbreak literature has not addressed this aspect to the authors' knowledge. The demonstrated complexity implies that efficient jailbreak optimization algorithms likely require exploitation of problem-specific structures. Our FH method, leveraging model fine-tuning, can be interpreted as utilizing the "alignment" property of models. A rigorous analysis of FH would necessitate an extensive exploration of neural network structures, LLM architectures, and typical language data characteristics, potentially advancing the frontiers of deep learning and LLM theory.

---

### Meta-Review · Area_Chair_DWUL · 2024-12-23

**Metareview:**

This paper proposes a new method called the Functional Homotopy (FH) method to solve discrete optimization problems that appear in jailbreak attacks on LLMs. The proposed method achieves 20%-30% improvement in success rates.

The proposed method is based on an interesting idea that is quite new in the literature. The method is nicely designed by solving a series of progressively harder problems one after another. Moreover, the performance improvement is significant.
The writing is clear and well organized. It is easy for readers to follow the context. The readers will benefit from reading this paper.

Hence, I recommend acceptance for this paper.

**Additional Comments On Reviewer Discussion:**

The reviewers were overall supportive on this paper. I agreed with their opinions and followed their evaluations.

---

### Decision · Program_Chairs · 2025-01-22

Accept (Poster)